# A national survey in United Arab Emirates on practice of passive range of motion by physiotherapists in intensive care unit

Gopala Krishna Alaparthi[1]*, Veena Raigangar[1], Kalyana Chakravarthy Bairapareddy[1], Aishwarya Gatty[2], Shamma Mohammad[1], Asma Alzarooni[1], Marah Atef[1], Rawan Abdulrahman[1], Sara Redha[1], Aisha Rashid[1], May Tamim[1]

1 Department of Physiotherapy, College of Health Sciences, University of Sharjah, Sharjah, United Arab Emirates, 2 College of Physiotherapy, Srinivas University, Mangaluru, Karnataka, India

* galaparthi@sharjah.ac.ae, gopalalaparthi@gmail.com

## Abstract

### Background

Patients admitted to intensive care units (ICU) are at an increased risk of developing immobility related complications. Physiotherapists are challenged to employ preventive and rehabilitative strategies to combat these effects. Passive limb range of motion (PROM) exercises- a part of early mobilization-aid in maintaining joint range of motion and functional muscle strength and forms a part of treatment for patients in ICU. However, there is a lack of evidence on practice of PROM exercises on patients admitted to ICU in the United Arab Emirates (UAE). This study aimed at exploring practices regarding the same in UAE.

### Methods

This survey, conducted from January 2021 to February 2021 in College of Physiotherapy, Sharjah University studied practice of physiotherapists in the intensive care units. Physiotherapists currently working in ICU completed an online questionnaire composed of forty-two questions about physiotherapy service provision, assessment and intervention in the intensive care units.

### Results

33 physiotherapists completed the survey. 66.6% of respondents routinely assessed PROM for all the patients in ICU referred for physiotherapy. 84.8% of them assessed all the joints. More than half of the respondents (57.8%) reported that they administered PROM regularly to all the patients. According to 63.6% respondents, maintaining joint range of motion was the main reason for performing PROM. Responses pertaining to sets and repetitions of PROM were variable ranging from 1–6 sets and from 3 to 30 repetitions. Personal experience, resources/financial consideration and research findings were found to have influence on the practice.

**Data Availability Statement:** All relevant data are within the manuscript and its Supporting Information files.

**Funding:** The authors received no specific funding for this work.

**Competing interests:** The authors have declared that no competing interests exist.

## Conclusions

PROM was found to be one of the frequently used mobilization techniques administered by physiotherapists in the intensive care units and was mostly performed after assessment. Maintaining joint range of motion was the main aim for performing PROM. Variability was found in the sets and repetitions of PROM administered. Various factors influenced the practice of PROM.

## Introduction

Patients with a life-threatening disease or trauma are admitted to the intensive care unit (ICU) with different stay durations from hours to months depending on the pathophysiology of the condition and the responsiveness to the given treatment [1,2]. They are usually confined to bed, which could have a negative impact on their mobility [3]. Prolonged immobilization, mechanical ventilation, and sedation have been associated with restricted joint mobility, critical illness neuropathies or ICU-acquired weakness, pressure sores, deep vein thrombosis (DVT), long duration of mechanical ventilation and cognitive impairments [3,4].

Continued inactivity and bed rest lead to decline in the use of skeletal muscles that result in reduced muscle synthesis and increased breakdown of protein with subsequent catabolism, weakness, and atrophy of muscles [5]. Myosin isoforms change from slow twitch to fast twitch fibers and metabolism changes from fatty acid to glucose [6]. Loss of muscle mass can cause up to 40% reduction in muscle strength within the first week of immobilization [5]. Critical illness survivors usually experience residual problems such as decreased physical function and mobility [7]. These sequelae have a great influence on the patients' overall functional activity and health-related quality of life [8].

Early mobilization is the early application and intensification of physical rehabilitation given to patients with critical illness, commenced within the initial two to five days of critical illness [9]. It includes activities such as in-bed mobility activities, range of motion exercises, sitting, standing, transfers, and gait training [6]. Early mobility in the ICU has been proposed to limit or prevent physical and cognitive dysfunction and provide various benefits [9,10]. Physiotherapists in ICU have an essential role in designing and practicing rehabilitation programs that aim to reinforce the mobility and strength of critically ill patients [11].

Ranges of motion exercises- active, active assisted and passive- maintain the muscle and joint integrity thus maintaining the function [12]. Passive range of motion is produced by an external force during muscular inactivity or when muscular activity is voluntarily reduced as much as possible to permit movements. Benefits of passive range of motion exercises include preventing adhesion formation and maintaining present free range of motion. When active movement is impossible, because of muscular insufficiency, these movements may help to preserve the memory of the movement patterns by stimulating the receptors of kinaesthetic sense. Extensibility of muscle is maintained and adaptive shortening is prevented. Venous and lymphatic return may be assisted slightly by mechanical pressure and by stretching of the thin walled vessels that pass across the joint moved [12].

A study was conducted in the United Kingdom to investigate the current use of passive joint movements by a physiotherapist working with sedated and ventilated patients in critical care settings. A questionnaire was distributed in England, Northern Ireland, Scotland, and Wales, among 246 physiotherapists working in ICU. The study reported that 99% of the respondents' practiced passive movements regularly in more than 70% of the patients admitted in ICU with medical, surgical or neurological disorders [11].

Likewise, a study in Australia was carried out to investigate physiotherapists' practice of passive limb range of motion on adult patients in ICU. A questionnaire was sent to physiotherapists working in level 3 adult ICU and responses showed variable application of passive limb range of motion. 35% of respondents undertook a routine assessment of passive limb range of motion for the patients admitted to ICU and 14% provided passive limb range of motion exercises in routine bases for all adult patients in ICU [8].

Variability in the practice of passive limb range of motion has been seen in different regions [8,11]. However, there is a lack of evidence on practice of passive limb range of motion in patients admitted to the ICU in the United Arab Emirates. Understanding the current practices may help in implementing and reforming its application in these patients and may also serve as a platform for future research. Therefore, this study aimed to investigate the current physiotherapy practice patterns regarding passive limb range of motion in the patients admitted to the ICU in the United Arab Emirates.

## Materials and methods

### Ethical considerations, registration of the study protocol and development of questionnaire

This survey, conducted from January 2021 to February 2021 in College of Physiotherapy, Sharjah University, studied practice of physiotherapists in the intensive care units. Approval was taken from the Research Ethics Committee, University of Sharjah (REC-21-01-S). A questionnaire was developed with reference to a previous survey done in Australia after taking consent from the authors [8]. The clinical practice of passive limb range of motion by a physiotherapist in ICU' questionnaire had forty-two questions that were divided among five main sections: physiotherapy service provision, physiotherapy assessment, and physiotherapy intervention, reflection on current practice, background, and personal information. It also consisted descriptive information regarding the ICU the participant worked in and other questions regarding the participant's characteristics. Some questions were close-ended, whereas others required ranking of responses using a scale. The questionnaire was given to five physiotherapists in ICU and an expert in the field of cardio-pulmonary physiotherapy for content validation (content validity ratio, CVR = 1).

### Selection of participants

Physiotherapists currently treating patients admitted to the intensive care units in the United Arab Emirates were eligible to be included in the survey. Those having less than one-year experience in treating critically ill patients were excluded.

### Recruitment

Lists of major private and governmental hospitals were obtained from the database of the Emirates physiotherapy society, UAE. Each hospital was contacted through physical visits or phone calls to identify the number of physiotherapists working in the Intensive Care Unit. From these hospitals (both government and private), 54 physiotherapists were identified as working in ICUs.

### Administration of questionnaire

The data collection took place over a period of four weeks. A Google hyperlink- consisting of a questionnaire along with a cover letter explaining the purpose of the study and a consent form- was sent to the 54 physiotherapists via email. The participants were requested to

complete the questionnaire based on their clinical experiences with the patients in ICU. Two weeks, from the date of mailing, was given for sending back the responses. A reminder was sent to non-responders after two weeks via phone calls and emails. The investigators waited for responses for another two weeks, after which the non-responder were excluded from the study.

## Statistical analysis

Descriptive summaries, frequencies and percetages were obtained by numerical coding of data and analysis using Statistical packages for social sciences (SPSS) version 16.36.

## Results

Questionnaires were sent to 54 participants out of whom 33 of them responded giving a response rate of 61.1%. Respondent characteristics and descriptive information regarding the intensive care units in which they practiced are shown in Table 1.

### Physiotherapy service provision

18 (54.5%) of the respondents indicated that their intensive care unit had a blanket referral for physiotherapy (all the patients admitted to the ICU are automatically referred for physiotherapy), whereas the remaining 15 (45.5%) of them required referral from medical staff. The mean and standard deviation of full-time equivalent (FTE) physiotherapists allocated to each intensive care unit for its weekdays regular service was 3.2±1.8, and the range was variable (1.0–8.0 FTE).

### Physiotherapy assessment

Most of the respondents (n = 22; 66.6%) indicated that all referred patients in the ICU were assessed by physiotherapists routinely. For respondents who reported that patients were not assessed routinely (n = 11; 33.3%), criteria for assessment included physiotherapists' judgment based on patients' medical records (n = 4; 12.1%) or current medical condition of the patients (n = 7; 21.2%).

A majority of the respondents assessed PROM routinely for all the patients in ICU (n = 25; 75.8%).For those not performing the assessment regularly, criteria for assessment included prolonged ICU length of stay (n = 5; 15.1%), the reason for admission (n = 6; 18.2%), past medical history (n = 3; 9.1%), patient sedation (n = 2; 6.1%), and intubation status (n = 1; 3.0%). Moreover, when assessing PROM, most respondents (n = 28; 84.8%) assessed all joints, while others assessed selected joints, as shown in Fig 1. 26 respondents (78.8%) reported that they used visual estimation measures to joint range of motions whereas others used goniometric measures (n = 6; 18.2%). 1 participant responded that he used objective assessment but the method was not specified. As shown in Fig 2, passive limb range of motion, for every patient, was most commonly assessed twice to three times per week (n = 15; 45.5%) and the assessment most often took 6–15 minutes per patient (n = 23; 69.7%).

### Physiotherapy intervention

Nineteen (57.8%) respondents indicated that they routinely treated all the patients with a passive limb range of motion exercises irrespective of their assessment findings. For the others, criteria for its administration, were as following: increased risk of loss of limb range of motion (e.g., increased tone, burns) (n = 11; 33.3%), a unilateral reduction in limb range of motion in

**Table 1. Characteristics and descriptive data of respondents and intensive care units.**

| Respondents | |
|---|---|
| **Age, year n (%)** | |
| 20–30 years | 22(66.7%) |
| 31–40 years | 6(18.1%) |
| 41–50 years | 4(12.1%) |
| >50 years | 1(3.0%) |
| **Sex (male/female) n (%)** | 12(36.4%)/21(63.6%) |
| **Years since graduation (Mean±SD)** | 7.4±7.7 |
| **Postgraduate qualification* n (%)** | |
| Master's Degree in Physiotherapy | 4(12.1%) |
| Doctor of Physiotherapy (DPT) | 1(3.0%) |
| Doctor of Philosophy (PhD) | 2(6.1%) |
| Sport physiotherapy certificate | 1(3.0%) |
| Master of Business Administration | 1(3.0%) |
| **Years of ICU experience n (%)** | |
| <5 years | 25(75.7%) |
| 5–10 years | 5(15.2%) |
| 11–15 years | 2(6.1%) |
| >15 years | 1(3.0%) |
| **Intensive Care Units** | |
| **No. of beds n (%)** | |
| <15 beds | 14(42.4%) |
| 16–30 beds | 11(33.3%) |
| 31–50 beds | 6(18.2%) |
| >50 bed | 2(6.1%) |
| **No. of admissions per year n (%)** | |
| <50 admission | 7(21.2%) |
| 50–100 admission | 19(57.6%) |
| >100 admission | 7(21.2%) |
| **Average duration of ICU stay in days (Mean±SD)** | 15.06±10.1 |
| **Average duration of mechanical ventilation in days** | 7.5±4.82 |
| **Types of patients admitted to ICU n(%)** | |
| Medical | 23(69.7%) |
| Surgical | 20(60.6%) |
| Trauma | 23(69.7%) |
| Burns | 17 (51.5%) |

*All physiotherapists practicing in United Arab Emirates are qualified with an undergraduate physiotherapy degree.

comparison to the patient's other side (n = 4; 12.1%), and reduced limb range of motion with respect to normal parameters (n = 6; 18.2%).

Tables 2 and 3 show the respondents' rankings based on their aim of performing passive limb range of motion exercises on the patients and the treatment technique/modalities used. Considering the aim of the treatment, maintaining joint range of motion was ranked as most important (n = 21; 63.6%) (Table 2). The most important treatment technique was manually applied passive limb range of motion (Table 3). 15 (45.5%) respondents reported that the aim of treatment affected their choice of treatment technique or modalities.

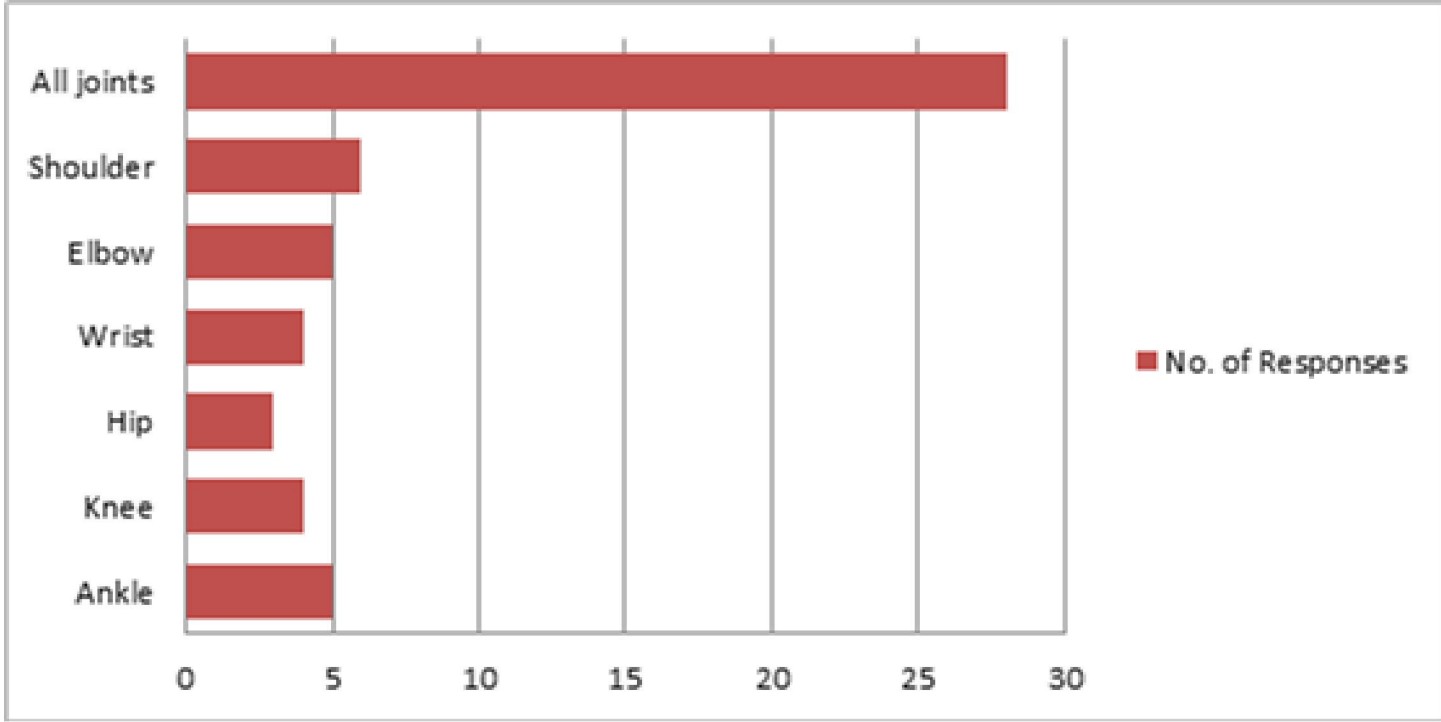

**Fig 1. Joints assessed for passive limb range of motion reported by the 33 respondents.**

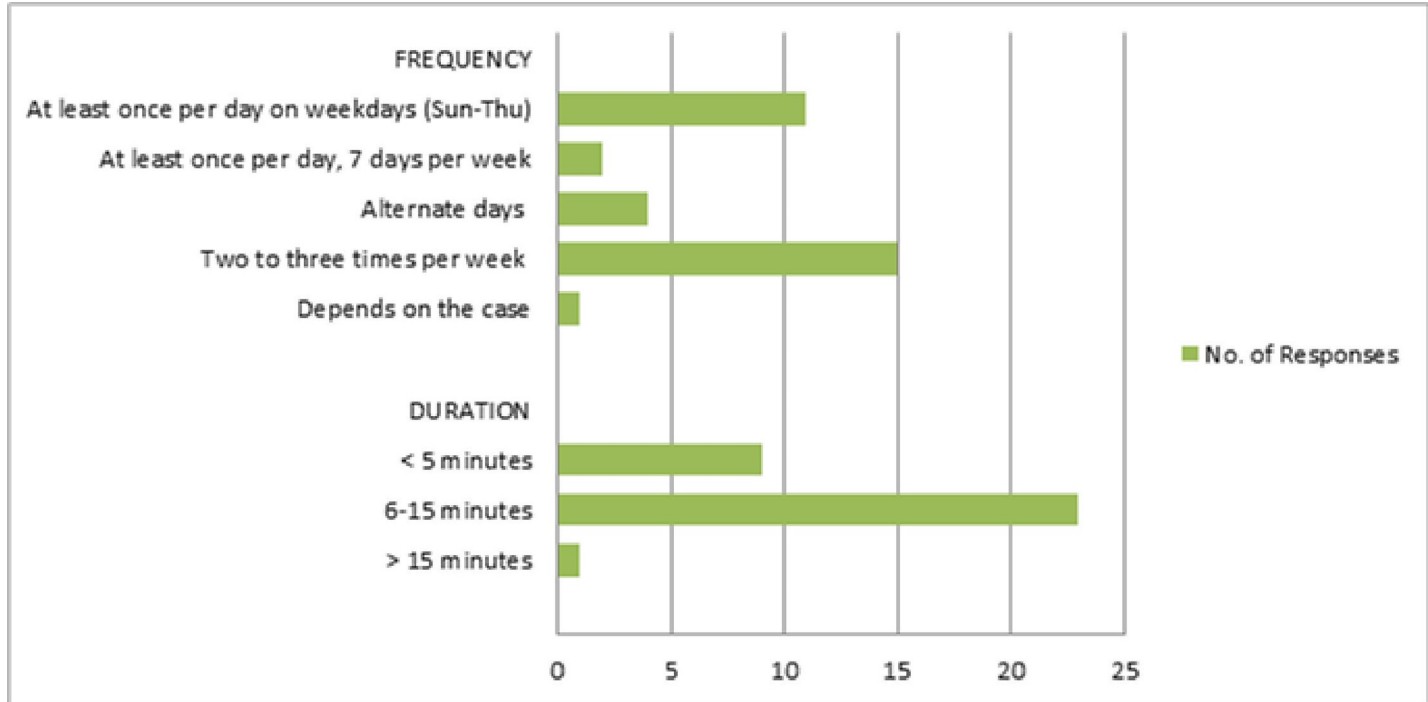

**Fig 2. Frequency and duration of performing passive range of motion assessment reported by the 33 respondents.**

**Table 2. Rankings awarded by respondents based on the importance of the aims of passive limb range of motion exercises.**

| Ranking Aims | 1 Most Important | 2 | 3 | 4 | 5 | 6 | 7 Least Important |
|---|---|---|---|---|---|---|---|
| Maintain joint range of motion | 21 | 5 | 3 | 0 | 0 | 3 | 1 |
| Maintain soft tissue extensibility | 1 | 19 | 4 | 4 | 1 | 2 | 2 |
| Reduce loss of joint range of motion | 1 | 0 | 17 | 6 | 6 | 1 | 2 |
| Reduce loss of soft tissue extensibility | 0 | 2 | 3 | 18 | 6 | 4 | 0 |
| Increase joint range of movement | 1 | 0 | 2 | 4 | 17 | 4 | 5 |
| Increase soft tissue extensibility | 2 | 3 | 2 | 1 | 1 | 18 | 6 |
| Preserve function | 7 | 4 | 2 | 0 | 2 | 1 | 17 |

Numbers in the boxes represent the frequencies of respondents.

Blue-0% of respondents.

White-1-25% of respondents.

Orange-26-50% of respondents.

Green-51-75% of respondents.

Yellow->75% of respondents.

Furthermore, highest number of respondents reported that they performed passive limb range of motion exercises once daily in weekdays i.e. Sunday to Thursday (n = 12; 36.4%). The second most common response was twice daily during the weekdays (n = 8; 24.2%) 1 respondent reported that its frequency depends on the case (condition of the patient). Responses pertaining to sets and repetitions of passive limb range of motion exercises were variable ranging from 1–6 sets and from 3 to 30 repetitions. Some of the respondents (n = 12; 36.4%) prescribed 3 sets of 10 repetitions while the others (n = 6; 18.2%) prescribed 1 set of 10 repetitions. End of resistance (n = 16; 48.5%) and limit of pain (n = 15; 45.5%) were the most common parameters, reported by the respondents, that were used to limit the range while administering these exercises. 26 (78.8%) of the respondents reported that they reassessed effects of passive limb range of motion and was mostly done via visual estimation (n = 26; 78.8%).

17(51.6%) respondents reported that nursing staff (n = 11; 33.3%) and physiotherapy assistants (n = 10; 30.3%), were the staff other than physiotherapists, who performed passive limb range of motion for patients in ICU. They followed instructions of physiotherapists (n = 10;

**Table 3. Rankings awarded by respondents based on the importance of the use of passive range of motion techniques/modalities.**

| Rankings Uses | 1 Most Important | 2 | 3 | 4 | 5 | 6 | 7 Least Important |
|---|---|---|---|---|---|---|---|
| Manually applied passive limb range of motion | 25 | 2 | 0 | 1 | 1 | 0 | 4 |
| Orthoses/splints | 0 | 13 | 7 | 5 | 4 | 2 | 2 |
| Positioning regimen | 1 | 9 | 13 | 2 | 4 | 4 | 0 |
| Continuous passive motion machine | 1 | 1 | 3 | 15 | 6 | 4 | 3 |
| Neuromuscular electrical stimulation | 1 | 3 | 4 | 5 | 11 | 4 | 5 |
| Mobilization (standing, walking) | 2 | 3 | 4 | 1 | 5 | 16 | 2 |
| Compression garments | 3 | 2 | 2 | 4 | 2 | 3 | 17 |

Numbers in the boxes represent the frequencies of respondents.

Blue-0% of respondents.

White-1-25% of respondents.

Orange-26-50% of respondents.

Green-51-75% of respondents.

Yellow->75% of respondents.

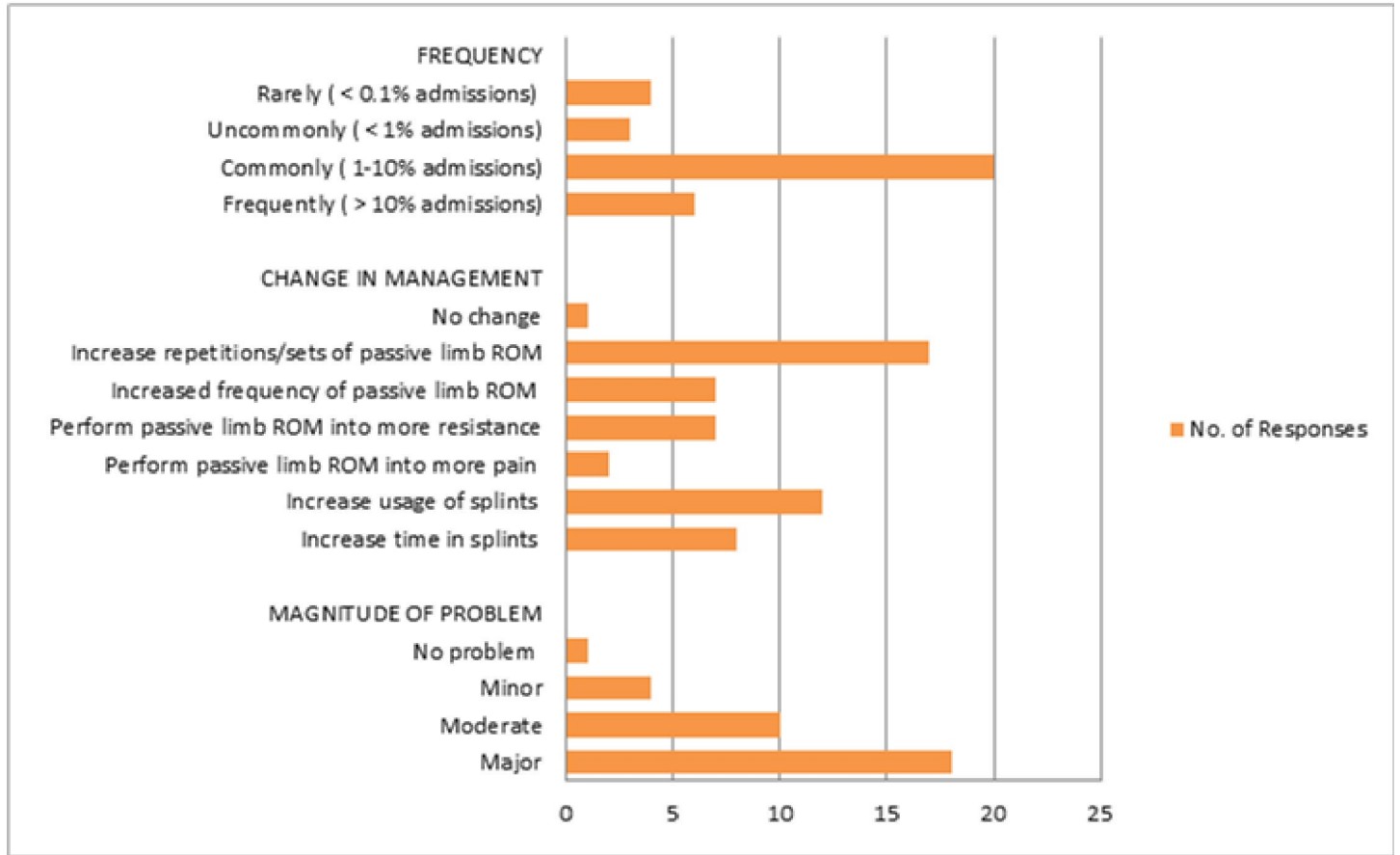

**Fig 3. Frequencies for the incidence, magnitude of the problem, and management in relation to loss of passive limb range of motion/contracture reported by the 33 respondents.**

30.3%), nursing staff (n = 5; 15.1%) or medical staff (n = 3; 9.1%). 11(33.3%) respondents indicated that all the patients were treated by the non- physiotherapist staff, 3(9.09%) of them indicated that only patients with a reduction in passive limb range of motion with respect to normal parameters were treated while the and the other 3(9.09%) of them reported that patients who were at increased risk of limb loss of range of motion were treated by them.

## Reflection of current practice

When reflecting on the current practice in intensive care units (Fig 3), most respondents (n = 20; 60.7%) commonly encountered patients with loss of passive limb range of motion or contractures (1–10% admissions) and were managed mostly by increased repetitions or sets of passive limb range of motion exercises (n = 17; 51.5%) or by increased use of splint (n = 10; 36.4%). According to 18(54.5%) respondents, loss of passive limb range of motion or contracture was a major problem. Patients' quality of life (n = 10; 30.3%) was ranked the highest and increased physiotherapy time required (n = 11; 33.3%) was ranked the lowest among the problems caused by the contractures (Table 4).

Some respondents (n = 12; 36.4%) indicated that they spent less than 25% of their time with intensive care unit patients doing passive limb range of motion exercises, while the others (n = 12; 36.4%) spent 25–50% of their time. 19 (57.6%) of them thought that the current

**Table 4. Rankings awarded by respondents for the problems that resulted from loss of passive limb range of motion/contracture.**

| Rankings Problems | 1 Most Important | 2 | 3 | 4 | 5 | 6 Least Important |
|---|---|---|---|---|---|---|
| Patient cosmesis | 9 | 0 | 4 | 9 | 4 | 7 |
| Patient hygiene | 3 | 11 | 8 | 3 | 6 | 2 |
| Patient function | 7 | 9 | 10 | 2 | 2 | 3 |
| Patient quality of life | 10 | 6 | 5 | 8 | 3 | 1 |
| Increased hospital length of stay | 2 | 5 | 2 | 6 | 9 | 9 |
| Increased physiotherapy time required | 2 | 2 | 4 | 5 | 9 | 11 |

Numbers in the boxes represent the frequencies of respondents.

Blue-0% of respondents.

White-1-25% of respondents.

Orange-26-50% of respondents.

Green-51-75% of respondents.

Yellow->75% of respondents.

practices with respect to these exercises were mostly effective. Although all the other factors were considered as predisposing factors for developing contractures, neurological conditions (n = 23; 69.7%) and burns (n = 23; 69.7%), were considered the most important factors (Fig 4).

Table 5 shows the ranking of factors that influence the physiotherapy practice with respect to the passive limb range of motion for the patients admitted to ICU. Personal experience, resources/financial consideration and research findings were found to have influence on the practice, whereas patients' status being public or private had the least influence.

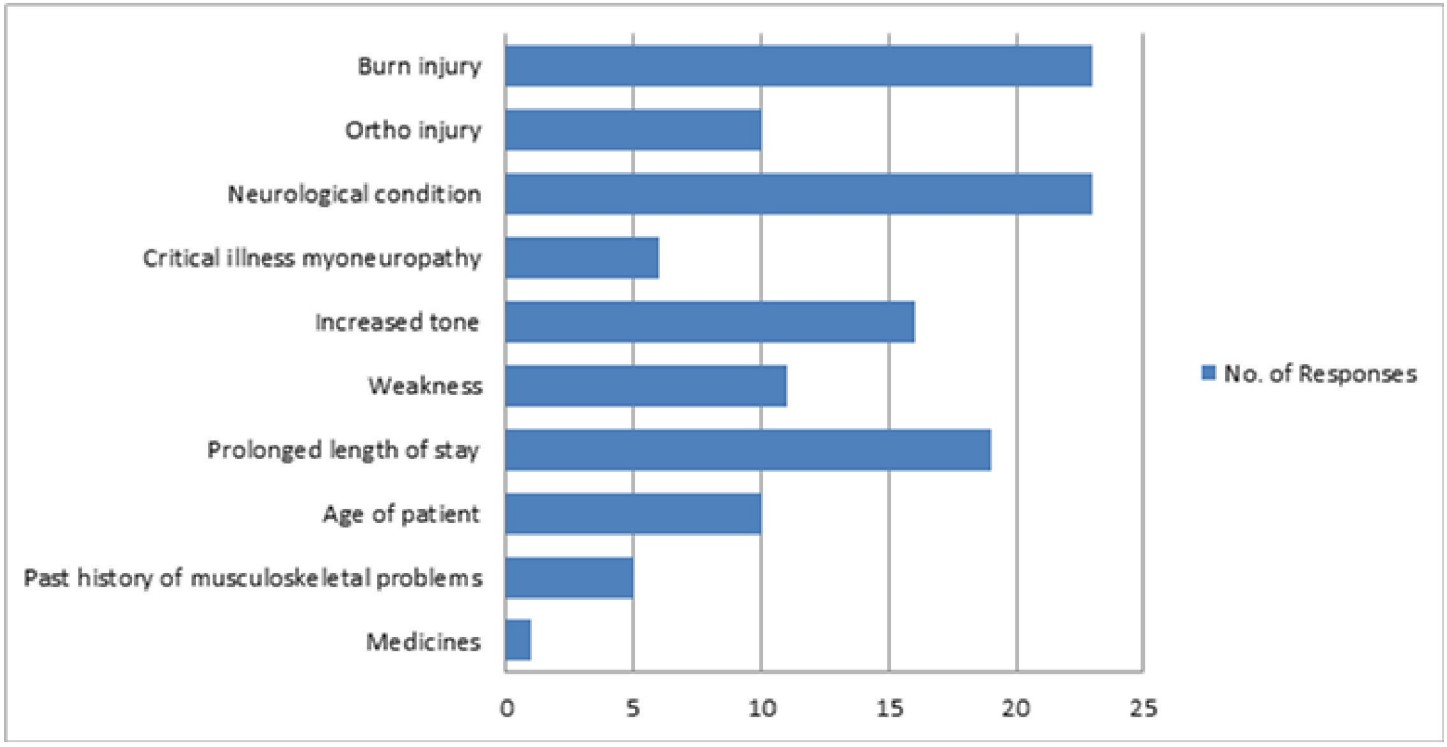

**Fig 4. Frequencies for predisposing factors for loss of passive limb range of motion reported by the 33 respondents.**

**Table 5. Rankings awarded by respondents for the influence of selected factors on respondents' physiotherapy practice with respect to passive limb range of motion.**

| Rankings Factors | 1 No Influence | 2 | 3 Moderate Influence | 4 | 5 Very Influential |
|---|---|---|---|---|---|
| Personal experience | 4 | 0 | 6 | 8 | 15 |
| Research findings | 1 | 7 | 10 | 9 | 6 |
| Advice from colleagues | 4 | 6 | 10 | 9 | 4 |
| Medical staff preferences | 4 | 8 | 8 | 11 | 2 |
| Resource/financial considerations | 4 | 7 | 8 | 7 | 7 |
| Established practice | 4 | 6 | 10 | 8 | 5 |
| Staffing numbers/caseload | 5 | 6 | 11 | 7 | 4 |
| Local/non-local patients | 15 | 5 | 7 | 4 | 2 |

Numbers in the boxes represent the frequencies of respondents.

Blue-0% of respondents.

White-1-25% of respondents.

Orange-26-50% of respondents.

Green-51-75% of respondents.

Yellow->75% of respondents.

## Discussion

This survey studied the practice of passive limb range of motion performed by physiotherapists in the intensive care units across the United Arab Emirates. We found that patients in the ICU received passive limb range of motion either by blanket referral (when all the patients admitted to the ICU are automatically referred for physiotherapy), or by referral from medical staff (doctors). A major portion of the respondents assessed passive limb range of motion routinely.

Range of motion can be measured using different methods including various goniometers or visual estimation [13]. Most respondents assessed all joints (shoulder, elbow, wrist, hip, knee, and ankle) routinely, usually by visual estimation. They assessed the passive limb range of motion twice or thrice a week, each assessment session taking a maximum 6 to 15 minutes.

Immobility, contraindication to active movements and stiffness are some of the indications for passive range of motion exercises [14]. But, our results showed that nearly half of the respondents provided these exercises for all the patients in ICU. Only a minority of respondents provided the treatment based on the patients' risk of developing limb range of motion loss.

Passive limb range of motion could be provided to patients manually or using equipment like continuous passive motion machines or cycle ergometer [15,16]. In our survey, the most frequently used mode of treatment by physiotherapists was manually applied passive limb range of motion exercises. Number of sets varied significantly among different respondents. Aims for performing these exercises are maintaining muscle strength and range of motion, minimizing contractures and assisting circulation [12,14]. We found that most of the respondents administered these exercises with the main aim of maintaining joint range of motion. A survey done in the United Kingdom also identified 'maintaining joint range of motion' and 'preventing contractures' as the main aims of administering passive range of motion to the patients in ICU [17].

Passive limb range of motion exercises can be administered to patients in ICU to increase proportion of perfused vessels [18] and strength of the muscles [19,20]. They can also decrease the pain, cytokine levels [21] and incidence of ICU acquired weakness [22] without having any significant hemodynamic changes, even in mechanically ventilated patients [16,18,21].

The common frequency of performing passive joint range of motion was found to be once daily, 5 days a week. The sets and repetitions had a varied range from 1–6 sets and from 3–10 repetitions for each joint. In a survey done in Australia, by Wiles et al, the common frequencies were found to be once daily for seven days a week and once daily for five days a week [8]. Similarly, the sets and repetitions had a variable range from 1–4 sets and 2 to 30 receptions for each joint. However, for patients who were developing or had already developed loss of joint range of motion, therapists intensified their interventions by adding sets and repetitions to the range of motion exercise and increasing the time in splints.

The survey reported that passive limb range of motion exercises were also performed by staff other than physiotherapists like nursing staff and physiotherapy assistants. Similar result was found in a study done by Wiles et al [8]. According to most of the respondents, time spent on these exercises was upto 50% of their time with the patients. Similar results found in the survey done by Wiles et al [8].

Joint contracture is a limitation in the passive limb range of motion of a joint secondary to shortening of the periarticular connective tissues and muscles. Abnormal posture, immobility and muscle weakness are predisposing factors for contractures [23]. In our survey, burns and neurological conditions were found to be the most common predisposing factor for contractures. Many respondents indicated that contractures were fairly common among the patients in ICU and that they were a major problem as the patients' cosmesis and quality of life would be affected.

We limited our study sample to physiotherapists only, as they are mostly responsible for mobility of the patients. Other studies mostly included senior physiotherapists (depending on the number of years of experience) as respondents [8,11]. Instead, we gathered responses from physiotherapists with at least one year of experience in the ICU to have an idea about overall practice of passive limb range of motion in patients admitted to ICU. Most of our respondents (66.7%) were juniors (20–30 years) and they considered experience as a factor that would influence their practice. The practice patterns explored in the study may be influenced by therapists' age, gender qualifications and years of experience.

The limitation of this survey was the small sample size, the reason being that there are less number of physiotherapists treating patients in ICU and having atleast one year of experience. Another factor that contributed to the small sample size was the response rate. This could be because this survey took approximately twenty minutes which could have led to reduction in completion rates and abandonment of survey. Also, this survey was conducted through online mode. Although it allows reaching a larger population in a short span of time, some participants- such as the senior population who are less familiar with advancement in technology- could have faced difficulty in filling the survey. This could be one of the reasons why most of our respondents were younger physiotherapists.

Our study is the first of its kind in the United Arab Emirates that provides descriptive data about the passive limb range of motion in ICU settings. Similar studies could take place in other countries of the Gulf Cooperation Council region to identify differences and allow the comparison of the different practices in these regions. The results of this study aided in understanding the practices of passive limb range of motion in ICU, which could guide in making protocols for the same. Physiotherapy practice in the intensive care units of the United Arab Emirates is still nascent and requires development in many domains. Studies of this nature help to highlight the practices of physiotherapists working in intensive care units and can aid in reforming these practices.

## Supporting information

**S1 Questionnaire.**
(DOC)

**S1 Data.**
(XLSX)

## Acknowledgments

The authors would like to thank Louise Wiles B and Kathy Stiller- authors of the study 'Passive limb movements for patients in an intensive care unit: A survey of physiotherapy practice in Australia' for permitting us to use their questionnaire for our study. We also like to thank all the respondents who took their time out to participate in our survey.

## Author Contributions

**Conceptualization:** Gopala Krishna Alaparthi, Veena Raigangar, Kalyana Chakravarthy Bairapareddy, May Tamim.

**Data curation:** Gopala Krishna Alaparthi, Veena Raigangar, Kalyana Chakravarthy Bairapareddy, Shamma Mohammad, Asma Alzarooni, Marah Atef, Rawan Abdulrahman, Sara Redha, Aisha Rashid.

**Methodology:** Gopala Krishna Alaparthi, Veena Raigangar, Kalyana Chakravarthy Bairapareddy, Aishwarya Gatty, Shamma Mohammad, Asma Alzarooni, Marah Atef, Rawan Abdulrahman, Sara Redha, Aisha Rashid, May Tamim.

**Resources:** Gopala Krishna Alaparthi, Veena Raigangar, Aishwarya Gatty.

**Writing – original draft:** Gopala Krishna Alaparthi, Veena Raigangar, Kalyana Chakravarthy Bairapareddy, Aishwarya Gatty, Shamma Mohammad, Asma Alzarooni, Marah Atef, Rawan Abdulrahman, Sara Redha, Aisha Rashid, May Tamim.

**Writing – review & editing:** Gopala Krishna Alaparthi, Veena Raigangar, Kalyana Chakravarthy Bairapareddy, Aishwarya Gatty.

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
