## [Decision Letter · Decision Letter 0]

29 Apr 2021

PONE-D-21-07815

A national survey in United Arab Emirates on practice of passive range of motion by physiotherapists in intensive care unit

PLOS ONE

Dear Dr. Alaparthi,

Thank you for submitting your manuscript to PLOS ONE. After careful consideration, we feel that it has merit but does not fully meet PLOS ONE’s publication criteria as it currently stands. Therefore, we invite you to submit a revised version of the manuscript that addresses the points raised during the review process.

We look forward to receiving your revised manuscript.

Kind regards,

Walid Kamal Abdelbasset, Ph.D.

Academic Editor

PLOS ONE

Journal Requirements:

2. Please upload a copy of the questionnaire as a supplemental file.

Reviewers' comments:

Reviewer's Responses to Questions

**Comments to the Author**

1. Is the manuscript technically sound, and do the data support the conclusions?

Reviewer #1: No

Reviewer #2: Yes

2. Has the statistical analysis been performed appropriately and rigorously? 

Reviewer #1: No

Reviewer #2: No

3. Have the authors made all data underlying the findings in their manuscript fully available?

Reviewer #1: No

Reviewer #2: No

4. Is the manuscript presented in an intelligible fashion and written in standard English?

Reviewer #1: Yes

Reviewer #2: Yes

5. Review Comments to the Author

Reviewer #1: Reviewer comments:

Abstract:

1. The background of the study is too long and also not justified the research gap.

2. The methods section is missing the study design, study setting, and the study duration.

3. The results component should consist of the correlation between the passive movement and its improvement in joint range of motion.

4. The conclusion is not drawn on the basis of the results drawn.

Article

1. How come this study is differing from articles 9, 01, 11?

2. How come the UAE ethnicity and demographic characters are differed from England and Australia people?

3. Authors failed to find the research gap and its clinical significance.

4. Include the ethical committee name and its reference number.

5. Include the reliability and validity of the questionnaire used for this survey.

6. Include the name of the questionnaire.

7. Include the method used for finding the sample size?

8. Mention the statistical analysis performed for this survey.

9. The results component should consist of the correlation between the passive movement and its improvement in joint range of motion.

10. Mention the role of age, gender, educational qualification, years of experience etc… in physical therapy intervention.

11. The discussion part should define the mechanism, how passive movement improves patient’s symptoms in ICU with recent references.

12. Overall, this study is not scientifically strong and also technically week for publishing.

Reviewer #2: PLOS ONE REVIEW

Topic: A national survey in United Arab Emirates on practice of passive range of motion by physiotherapists in intensive care unit

Abstract

Line 33: Sample size extremely too small for a national survey.

Line 39: 51.6% is more than half and not almost half

Line 43: What assessment is being referred to? statement not clear.

Introduction

Line 78: Effects should be changed to Benefits

Materials and Methods

Line 110: was made should be changed to was developed.

Line 116: Reflection on their current practice is repeated. It is already written in line 113.

Lines 122-125: What are the exclusion criteria. This statement on inclusion criteria is not clear.

Lines 126-127: Was required sample size not calculated? What physiotherapy Groups exactly? This has to be clearly stated. Would the national licensing body not be better? Were they from both private and government hospitals?

Results

Line 140: Are there a total 54 ICU physiotherapist with at least one year experience?

Line 143: Table should be properly labelled as n (%), n, meaning frequency. The frequency data under age, gender and years of ICU experience do not add up to 33. The mean value for year of graduation says .3, I don’t know how that can be. The average duration of ICU stay should be clear if it was recorded in days, weeks or months. Under types of patients admitted to the ICU, Burns (the frequency and the percentage) was written twice. What is GCC Region? Please be clear on that. Generally, be consistent with the number of decimal points.

Line 148: What is Blanket referral? Please be clear on that.

Line 151: Please include the standard deviation to the mean of 3.75

Lines 166-168: It is good to clarify if the statement is per patient.

Line 182: Tables 2 and 3. S should be added to table.

Line 190 and 194: Tables should be better labelled to indicate that these figures are all frequencies. The coding is not clear.

Line 204: Percentage of 37.5 cannot be referred to as majority.

Line 206 and 207: The percentage calculation of 26 (81.3%) or 26 (96.3%) is not correct.

Line 212: Frequency of 10 cannot be referred to as most of the respondents. Frequency of 10 cannot give a percentage of 55,6.

Line 223: What percentage of the respondents reported contractures to be the major problem.

Line 238: Table should be better labelled to indicate that these figures are all frequencies. The coding should be made clearer.

Line 249: According to figure 4, we had both neurological condition and burns, can this be made clearer?

Line 260: Table should be better labelled to indicate that these figures are all frequencies. The coding should be made clearer.

Many of the percentage calculations are not correct. Some are lower, while some are higher.

It is difficult to ascertain the actual sample size, 30 0r 33. Please this has to be clear, what exactly is the sample size?

Discussion

Line 269: What is Blanket referral? Can it be reworded?

Line 270: What other staff? Doctors? other physiotherapist? It has to be clearly stated.

Line 315: Who are senior physiotherapist? By age or by cadre? Are senior Physiotherapists not included at all in this study.

Line 318-320: Could this be as a result of the mode of data collection? could it be that younger individuals generally have better reaction towards activities online?

Line 321: I am worried that the sample size of 33 or 30 may not represent the perception of ICU physiotherapists in the UAE.

What are the possible limitations to this study?

Any acknowledgements?

Figures

Can the bars on Fig 3 be better aligned?

Raw Data

Not available for review.

6. PLOS authors have the option to publish the peer review history of their article (what does this mean?). If published, this will include your full peer review and any attached files.

Reviewer #1: No

Reviewer #2: **Yes: **Ajepe, Titilope Oluwatobiloba

---

## [Author Response · Author response to Decision Letter 0]

10 Jun 2021

Thank You for the corrections. We have tried our best to incorporate all the corrections. Please find attached the revised manuscript and response to reviewers file for your kind perusal.

---

## [Editor Report · Decision Letter 1]

28 Jun 2021

PONE-D-21-07815R1

A national survey in United Arab Emirates on practice of passive range of motion by physiotherapists in intensive care unit

PLOS ONE

Dear Dr. Alaparthi,

Thank you for submitting your manuscript to PLOS ONE. After careful consideration, we feel that it has merit but does not fully meet PLOS ONE’s publication criteria as it currently stands. Therefore, we invite you to submit a revised version of the manuscript that addresses the points raised during the review process.

We look forward to receiving your revised manuscript.

Kind regards,

Walid Kamal Abdelbasset, Ph.D.

Academic Editor

PLOS ONE

Additional Editor Comments (if provided):

Five new authors have been added to the authors list.

Each authors’ contribution should be explained. Please provide the authors’ contributions in line with ICMJE4 criteria.

The major concern in the study is the small sample size compared with the study design. The authors have to explain how did they calculate the sample size and power of the study?

---

## [Author Response · Author response to Decision Letter 1]

5 Jul 2021

Thank You for your comments. We have provided explanation for your valuable comments.

---

## [Decision Letter · Decision Letter 2]

9 Aug 2021

A national survey in United Arab Emirates on practice of passive range of motion by physiotherapists in intensive care unit

PONE-D-21-07815R2

Dear Dr. Alaparthi,

We’re pleased to inform you that your manuscript has been judged scientifically suitable for publication and will be formally accepted for publication once it meets all outstanding technical requirements.

Kind regards,

Walid Kamal Abdelbasset, Ph.D.

Academic Editor

PLOS ONE

Additional Editor Comments (optional):

Reviewers' comments:

Reviewer's Responses to Questions

**Comments to the Author**

1. If the authors have adequately addressed your comments raised in a previous round of review and you feel that this manuscript is now acceptable for publication, you may indicate that here to bypass the “Comments to the Author” section, enter your conflict of interest statement in the “Confidential to Editor” section, and submit your "Accept" recommendation.

Reviewer #1: All comments have been addressed

Reviewer #3: All comments have been addressed

Reviewer #4: All comments have been addressed

2. Is the manuscript technically sound, and do the data support the conclusions?

Reviewer #1: No

Reviewer #3: Yes

Reviewer #4: Yes

3. Has the statistical analysis been performed appropriately and rigorously? 

Reviewer #1: I Don't Know

Reviewer #3: Yes

Reviewer #4: Yes

4. Have the authors made all data underlying the findings in their manuscript fully available?

Reviewer #1: Yes

Reviewer #3: Yes

Reviewer #4: Yes

5. Is the manuscript presented in an intelligible fashion and written in standard English?

Reviewer #1: Yes

Reviewer #3: Yes

Reviewer #4: Yes

6. Review Comments to the Author

Reviewer #1: Dear Author,

I regret to say that this is a very outdated title, which could not be reviewed.

Regards

Reviewer #3: -

Reviewer #4: Dear authors of the manuscript entitled "A national survey in United Arab Emirates on practice of passive range of motion by physiotherapists in intensive care unit

" i found your article quite intresting, i have no comments and i believe that the manuscript is eligible for publication

best wishes

7. PLOS authors have the option to publish the peer review history of their article (what does this mean?). If published, this will include your full peer review and any attached files.

Reviewer #1: No

Reviewer #3: No

Reviewer #4: No

---

## [Editor Report · Acceptance letter]

13 Aug 2021

PONE-D-21-07815R2 

A national survey in United Arab Emirates on practice of passive range of motion by physiotherapists in intensive care unit 

Dear Dr. Alaparthi:

I'm pleased to inform you that your manuscript has been deemed suitable for publication in PLOS ONE. Congratulations! Your manuscript is now with our production department. 

Kind regards, 

on behalf of

Dr. Walid Kamal Abdelbasset 

Academic Editor

PLOS ONE